# Can Primary Medical Institutions Lead to Worse Health Status for Patients with Noncommunicable Diseases Compared with High-Level Hospitals? A Follow-Up Observation Study in China

**DOI:** 10.3390/ijerph16081336

**Published:** 2019-04-14

**Authors:** Yadong Niu, Ting Ye, Yan Zhang, Liang Zhang

**Affiliations:** 1School of Medicine and Health Management, Tongji Medical College, Huazhong University of Science and Technology, Wuhan 430030, China; nyadong@126.com (Y.N.); yeting868@163.com (T.Y.); yanzhang@hust.edu.cn (Y.Z.); 2Research Centre for Rural Health Service, Key Research Institute of Humanities & Social Sciences of Hubei Provincial Department of Education, Wuhan 430030, China

**Keywords:** non-communicable diseases, health-related quality of life, service quality, primary medical institutions, China, follow-up study

## Abstract

The weak primary healthcare system in China brings challenges to the national strategy of primary medical institutions providing general health needs for patients with non-communicable diseases (NCDs). It is necessary to explore the potential discrepancies in health status for patients with NCDs if they go to primary medical institutions rather than high-level hospitals. Data was obtained from Surveillance of Health-seeking Behavior in Hubei Province. Respondents were investigated six times to collect information on health service utilization and health-related quality of life (HRQoL). Ninety-two hypertension patients who went to medical institutions of the same level were included. HRQoL was measured by the Chinese version of EQ-5D-3L. A multilevel growth curve model was applied to analyze whether provider level could influence HRQoL. The utility score and visual analogue scale (VAS) of patients varied insignificantly over six months (*p* > 0.05). A growth curve model showed that comorbidity was the only factor significantly influencing utility score (*p* = 0.019). Time and comorbidity were the only influencing factors of VAS (*p* < 0.05). Our findings indicated that the level of healthcare provider had no significant impact on the health status of patients with NCDs. As such, this study concludes that the primary healthcare system in China is qualified to be the health gatekeeper for NCDs patients.

## 1. Introduction

Non-communicable diseases (NCDs) have become the major health threat in low-, middle-, and high-income countries [1,2,3], and they account for more than 50% of the global disease burden [4,5]. NCDs are also responsible for 70% of deaths worldwide, and more than 40% of these deaths are premature [3]. Aging and changed lifestyle can make the status worse [6,7]. It is estimated that in 2030, over three-quarters of all deaths will be caused by NCDs [1,3]. NCDs can not only influence an individual’s life but can also impede sustainable development [1,8,9,10], thereby attracting attention from high-level politicians [8,10]. The current status of NCDs in China is no better than that in other countries. According to the Fifth Chinese National Health Services Survey, conducted in 2013, the prevalence rate of NCDs in China is 24.5% (much higher than the 18.9% in 2008), and 78.4% of the aged (over 65 years) present at least one NCD [11]. The World Bank and World health Organization indicate that NCDs are responsible for approximately 70% of the total disease burden and 89% of deaths in China [1,12].

Patients with NCDs present a large demand for health services and health resources [6,13]. Under such circumstances, the Chinese government hopes that the patients with NCDs in stable or rehabilitation stages can be managed by primary medical institutions [14,15]. However, China has a weak primary healthcare system [16,17]. If patients with NCDs go to primary medical institutions for common health needs, it is currently unknown whether their health will suffer compared to those who go to high-level hospitals.

To date, many studies have been performed on factors influencing the health status of patients with NCDs [18,19,20,21,22,23,24,25,26]. These studies mainly focused on psychiatric condition [21,22,24,25], disease progression [18,20,21,22,24], comorbidities [23,25], health-risk behavior [19,26], and demographic characteristics [18,20,24,26]. Few studies have focused on the level of providers.

In this study, we selected patients who were seeking health care only in hospitals and those only in community health centers (or primary medical institution) to analyze the influence of provider level on the health status of patients with NCDs. The results of this study can ease the public’s worry on capability and service quality for primary medical institutions and bring support for health policy making in China.

## 2. Materials and Methods

### 2.1. Setting and Study Sample

This is a follow-up observation study. Data was obtained from the Surveillance of Health-seeking Behavior in Hubei province (the third stage), which was conducted by the Health and Family Planning Commission of Hubei Province. Participants were selected from the samples of the Fifth National Health Service Survey of China in 2013, which was based on multi-stage stratified cluster random sampling. A total of 330 households (936 people in total) in Qingshan District, Wuhan City, Hubei Province, were followed up. The surveillance was approved by all the sample families prior to the data collection, with the commitment that collected information will be confidential. Experienced general practitioners from the local community health centers were employed to collect information about health service utilization (including all the outpatient and inpatient services and corresponding institutions, diseases, and fees) from participants by household survey at the end of each month. Health-related quality of life (HRQoL) is a common measure to evaluate the effect of an intervention on patients’ health status [27]. It was measured by the Chinese version of EQ-5D-3L (EQ-5D) monthly; this measure is widely used and highly recommended [28,29,30]. Directors of community health centers were employed as supervisors to ensure there was no logical errors, wrong items, or absent items in the questionnaire. The study period was six months (July–December 2016), and each family was investigated six times. The baseline survey was conducted during the first interview in July, and information on demographics, socioeconomic characteristics, and health status was collected. 

Hypertension is a considerable risk factor of cardiovascular, diabetes, and other chronic diseases [31]. It shows the highest morbidity among NCDs in China (43.1% of patients with NCDs present hypertension), accounts for nearly 24% of outpatient services (the second highest), and 5.4% of inpatient services (the third highest) [11]. Therefore, we selected hypertension patients as study participants. Among the 936 respondents, 240 people were diagnosed with hypertension before the study period, 126 of them used outpatient services for hypertension or relative disease during the six-month study period, and 107 of them filled out the EQ-5D completely by themselves six times. We divided those 107 people into the following groups according to the institution they chose for outpatient services: group 1 only went to community health centers (49 people), group 2 only went to hospitals (43 people), and group 3 went to both community health centers and hospitals (15 people) (Figure 1). Groups 1 and 2 were study participants, and we compared the trajectories of their HRQoL over six months.

### 2.2. Measurement of HRQoL

The HRQoL of respondents was measured by EQ-5D. The scale contains a health description system and visual analogue scale (VAS). The description system presents five dimensions, including mobility, self-care, activity, pain/discomfort, and anxiety/depression. Detailed information about the EQ-5D is provided in the EQ-5D-3L manual [32]. The EQ-5D description system was converted into a utility score according to the Chinese population-based time trade-off model, which was established by Liu G et al. [33].

### 2.3. Statistical Analysis

The growth curve model introduced by Potthoff and Roy in 1964 has been widely used in repeated measurement studies of various fields [34,35,36,37,38]. The model exhibits higher levels of statistical power than those of traditional methods when analyzing repeated measures data, especially in social science research [39]. We applied a multilevel growth curve model to analyze whether the healthcare provider level can influence the HRQoL of hypertension patients. The utility score and VAS from EQ-5D were both taken as outcome variables. Time and healthcare provider level (community health center and hospital) were considered as independent variables. Other factors, including age (less than 65, 65–70, 70–75, and over 75), sex (male and female), family size (less than 3, 3, and over 3), family income per person (less than 20,000¥, 20,000–30,000¥, and over 30,000¥), frequency of use of outpatient services (less than once every two months and more than once every two months), and comorbidity (no comorbidity, one comorbidity, and more than one comorbidities) were also taken as independent variables, which may also influence the HRQoLf patients [18,19,20,21,22,23,24,25,26]. The model of this study is shown below:(1)$$Y_{\mathrm{ij}}=\pi_{0i}+\pi_{1i}\mathrm{Time}+\varepsilon_{\mathrm{ij}}$$
(2)$$\pi_{0i}=\beta_{00}+\beta_{01}Provider_{level}+\beta_{02}Age+\beta_{03}Sex+\beta_{04}Family_{size}+\beta_{05}Income\;+\beta_{06}Outpatient_{frequency}+\beta_{07}Comorbidity+\gamma_{0i}.\;$$
(3)$$\pi_{1i}=\beta_{10}+\gamma_{1i}$$

*Y_ij_* denotes the HRQoL of patient i at time of *j*. *π_0i_* denotes the predicted HRQoL of *i* at the first wave. *π_1i_* represents the estimated slope (the predicted change of the HRQoL per month for patient *i*). *ε_ij_* refers to the errors of the HRQoL for patient *i* at time of *j*, and *β_00_* represents the average HRQoL at wave 1, with other variables equal to zero. *β_01_* represents the differences in the mean HRQoL between patients who selected hospitals as their common care provider and patients who chose community health center as their common care provider. *β_03_* refers to differences in the mean HRQoL between males and females. *β_02_*, *β_04_*, *β_05_*, *β_06_*, and *β_07_*, represent the effects of age, family size, family income, frequency of use of outpatient services, and comorbidity number on patients’ HRQoL. *β_10_*refers to the average change of the HRQoL per month for all the patients. *γ_0i_* and *γ_1i_* denote the *π_0i_* and *π_1i_* errors.

One-way ANOVA was used to analyze the variation in HRQoL over time. All analyses were conducted in SAS, version 9.4 (SAS Institute Inc., Cary, NC, USA). The growth curve model was performed by SAS Proc Glimmix procedure, and parameters were estimated by the maximum likelihood method. Values with *p* < 0.05 (two-tailed) were considered statistically significant.

### 2.4. Ethics Approval and Consent to Participate

The study protocol conformed to the guidelines of the Ethics Committee of the Tongji Medical College of Huazhong University of Science and Technology. The protocol was registered in the Chinese Clinical Trial Registry (ChiCTR-OOR-14005563). Patient information was anonymized and de-identified before analysis.

## 3. Results

### 3.1. Summary Statistics of Sample Hypertension Patients

Among the 92 hypertension patients, 53.3% of them selected community health centers for outpatient services and the majority of them were over 65 years old (77.2%). Moreover, 51.1% of the patients were female, and the household income per capita was distributed mainly in the 20,000–30,000¥ income category (43.5%). More than half of the patients with hypertension belonged to households with one or two members (55.4%), and most of them sought outpatient services less than once every two months (58.7%); more than half of these patients had at least one comorbidity (51.1%). Further details are provided in Table 1.

### 3.2. HRQoL of Hypertension Patients Over Time

We used the utility score and VAS from the EQ-5D to represent the HRQoL of the sample patients. During the six-month study period, the variation in the utility score mean was small and insignificant (*p* = 0.93). The highest mean of the utility score was 0.928, and the lowest was 0.903. The mean VAS also varied insignificantly (*p* = 0.155). The highest mean of VAS was 74.152, and the lowest mean was 69.685. Details are presented in Table 2.

### 3.3. Factors Associated with HRQoL in Hypertension Patients

To explore the factors influencing the HRQoL of patients with hypertension, the growth curve model was applied, and results are provided in Table 3. Comorbidity was the only factor significantly influencing the utility score. A high number of comorbidities in patients with hypertension resulted in a low utility score (estimate = −0.048, *p* = 0.019). Other factors (time, provider level, age, sex, household income per capita, family size, and frequency of use of outpatient services) exerted no significant influence on the utility score. Time also notably influenced the VAS; VAS increased with time (estimate = 0.508, *p* = 0.04). Comorbidity also significantly influenced the VAS. A high number of comorbidities in patients with hypertension resulted in low VAS (estimate = −4.62, *p* < 0.001). Other factors (provider level, age, sex, household income per capita, family size, and frequency of use of outpatient services) also exerted no significant influence on VAS.

## 4. Discussion

This observational study focused on the influence of healthcare provider level on the HRQoL of patients with NCDs. The healthcare provider level for respondents was unknown at the beginning but determined by their health-seeking behavior information, which was continuously collected for months. This study is a longitudinal study, and each respondent was interviewed face-to-face for six times (once a month). The total sample is 552 (92 respondents with six waves of interviews) and the results are more stable and reliable than those of cross-sectional studies.

Study participants were mainly people aged over 65 years, which was in agreement with the fact that elderly people consist of the majority of patients with NCDs [11]. Although our sample disease was hypertension, more than 50% of the respondents also had other NCDs, which was also in accordance with the NCDs prevalence of China [40,41]. Those two characteristics improved the representativeness of the sample and ensured the reliability of results.

The results showed that the HRQoL of the sample patients varied little over 6 months, which indicated that their health status was stable. The average utility score of sample patients was between 0.9 to 0.92, which was lower than that of the general population [42,43], and in accordance with the results measured by Li Zhang et al. (with the same method) for patients with hypertension [31]. The average utility score further proved that the results were reliable.

The growth curve model showed that time exerted no influence on the factor of the utility score, which indicated that the utility score underwent no significant change over time. This result matched the result of a one-way ANOVA. Time was significantly related to VAS, and VAS increased over time. VAS refers health status evaluated by patients themselves, and it is more subjective than a utility score. The change in VAS may be due to psychological effects, given that the patients have been going to medical institutions frequently, and they may have expected good health status, although their body functions show no considerable improvement.

Comorbidity was also an influencing factor for the utility score and VAS; a high number of comorbidities in patients resulted in a low utility score and VAS, which was consistent with the findings of other studies [23,25]. Previous studies showed that age, income, and family size are related to the HRQoL of patients [18,26], and patients who were younger, richer, and/or with more household members had better HRQoL. However, in our study, these factors exerted no significant influence on HRQoL. This result may be attributed to the small sample size and short tracking period. If more respondents were followed up for a longer time, significant influences may be observed. Additionally, the frequency of use of outpatient services also showed insignificant influence, which indicated that patients may not obtain an improved health status despite the use of considerable health services.

The biggest concern of this study is the influence of healthcare provider level on patients’ HRQoL. The growth curve model showed that healthcare provider was insignificantly related to the utility score or VAS, which indicated that healthcare provider level exerted no influence on the health status of patients. This suggests that if hypertension patients go to primary medical institutions for their common heath needs, their health status will not decline; in other words, primary medical institutions can provide general health services for hypertension patients as good as hospitals. In addition, more than 50% of the participants presented other NCDs, which suggested that the results can also be applied to patients with multi-comorbidity. The results of this study indicate that the health status of patients with multi-comorbidity will not suffer if they choose primary medical institutions as their common healthcare provider.

## 5. Limitations

Our study has several limitations. The sample size was small and the tracking time was short, which may influence the stability of our conclusions. Only hypertension patients were included, and this may limit the generalizability of conclusions. Comorbidity cannot fully reflect disease severity for hypertension patients and may cause bias of conclusions. Further studies need to include more diseases and more participants with longer observation periods to enhance the stability and generalizability of conclusions. Additionally, more characteristics of participants need to be collected to reduce potential interruptions.

## 6. Conclusions

Our study shows that the healthcare provider level will not influence the health status of aged patients with NCDs. If aged patients with NCDs go to primary medical institutions for their common health needs, their health is unlikely to suffer. Patients with NCDs should be encouraged to go to primary medical institutions for general health services, and primary medical institutions should take care of the health demands of patients with NCDs at stable or rehabilitation stages of their disease management. If those measures can be well performed, the utilization efficiency of health resources would be greatly increased, and the operating efficiency of the delivery system will be markedly improved.

## Figures and Tables

**Figure 1 ijerph-16-01336-f001:**
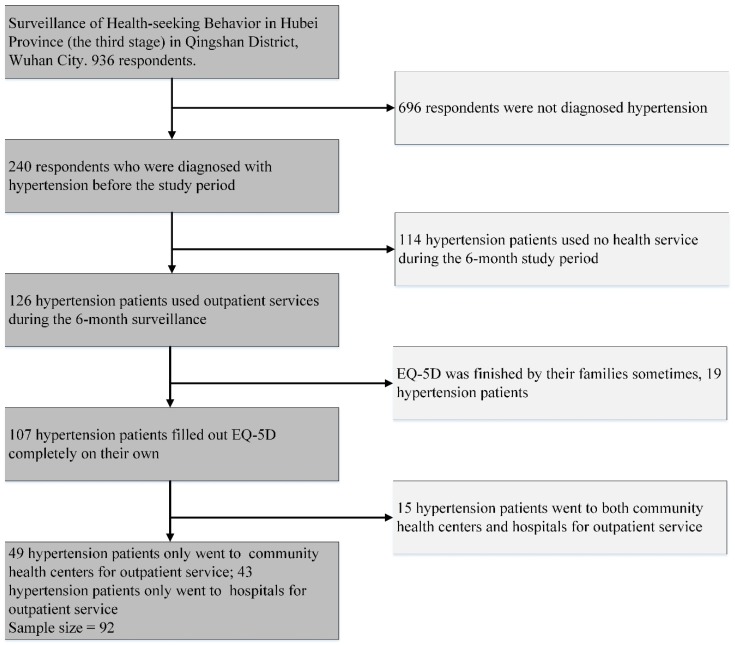
Procedure of sample selection.

**Table 1 ijerph-16-01336-t001:** Summary statistics of sample hypertension patients.

Variables	*n*	%
Provider level		
community health centers	49	53.3
hospitals	43	46.7
Age		
<65	21	22.8
65–70	19	20.7
70–75	23	25
>75	29	31.5
Sex		
male	45	48.9
female	47	51.1
Household income per capita		
<20,000¥	26	28.3
20,000–30,000¥	40	43.5
>30,000¥	26	28.3
Family size		
<3	51	55.4
3	25	27.2
>3	16	17.4
Frequency of use of outpatient services		
≤1 times/2 month	54	58.7
>1 times/2 month	38	41.3
Comorbidities		
0	45	48.9
1	22	23.9
>1	25	27.2

**Table 2 ijerph-16-01336-t002:** Health-related quality of life (HRQoL) of hypertension patients over time.

Month	Utility Score ^1^		VAS ^2^	
	mean	SD	mean	SD
July	0.903	0.185	69.685	14.305
August	0.920	0.166	74.152	11.367
September	0.928	0.146	72.957	13.213
October	0.912	0.152	74.130	11.402
November	0.915	0.151	72.891	12.635
December	0.921	0.148	73.761	13.129

Note: “1” *F* = 0.269, *p* = 0.93; “2” *F* = 1.611, *p* = 0.155. VAS: visual analogue scale.

**Table 3 ijerph-16-01336-t003:** Growth curve model of HRQoL.

	Utility Score			VAS		
	Estimate	SE	*p*	Estimate	SE	*p*
Intercept	1.001	0.063	<0.001	78.982	4.192	<0.001
Time	0.002	0.002	0.413	0.508	0.244	0.040
Provider level	−0.019	0.035	0.575	−1.788	2.294	0.436
Age	−0.016	0.015	0.260	−0.503	0.962	0.602
Sex	−0.034	0.032	0.294	−1.095	2.121	0.606
Household income per capita	−0.011	0.022	0.609	−1.475	1.456	0.312
Family size	0.016	0.021	0.450	1.657	1.407	0.240
Frequency of use of outpatient services	0.006	0.033	0.854	−3.730	2.177	0.088
Comorbidities	−0.048	0.019	0.011	−4.620	1.261	<0.001

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
