# Peer review of "Can Primary Medical Institutions Lead to Worse Health Status for Patients with Noncommunicable Diseases Compared with High-Level Hospitals? A Follow-Up Observation Study in China"

_ijerph, 2019, doi:10.3390/ijerph16081336_

Round 1
Reviewer 1 Report
65-67 This sentence reads as confusing.
75- How did the supervisors insure quality?
167- Instead of "old" consider the term "elderly."
197- "study objects" should be "study participants."
I believe that the authors did a nice job identifying the abilities and public confidence in primary care providers. My curiosity is in the definition of "hypertension" and the level of primary care use. Were the participants with mild hypertension more likely to seek primary care outside the hospital. While co-morbidity could signal this, it is less than direct.
Did hypertension also include those suffering hypertensive urgency or hypertensive emergency? How effective were primary care physicians in treating (or referring) these patients?
Author Response
Response to reviewers’ comments
ijerph-479573
Can Primary Medical Institutions Lead to Worse Health Status for Patients with Noncommunicable Diseases Compared with High-Level Hospitals? a Follow-up Observation Study in China
Dear editor and reviewer,
Thanks for your incredible work for reviewing our manuscript. Firstly we would like to express our gratitude to you for your valuable input. Secondly we have made some revisions in accordance with your comments and advices, and would like to re-submit the revised manuscript.
Our responses to your comments are below:
1. 65-67 This sentence reads as confusing.
Response: line 65-67 introduced the data source of the manuscript. Our data was obtained from a survey named “Surveillance of health-seeking behavior in Hubei province (the 3rd stage)”. The surveillance was carried out by the Health and Family Planning Commission of Hubei Province. Participants of the surveillance was selected from the Fifth National Health Service Survey of China in 2013, which was a Multi-stage stratified cluster random sampling.
We have revised line 65-67 as follows:
“Data were obtained from the Surveillance of Health-seeking Behavior in Hubei province (the 3rd Stage), which was conducted by the Health and Family Planning Commission of Hubei Province. Participants were selected from the samples of the Fifth National Health Service Survey of China in 2013, which was a Multi-stage stratified cluster random sampling. A total of 330 households (936 people in total) in Qingshan District, Wuhan City, Hubei Province were followed-up.” (Line 58-62)
2. 75- How did the supervisors insure quality?
Response: Directors of community health centers were employed to make sure that there was no Logical error, wrong item or absent item in the questionnaire. We have added more explanation.
“Directors of community health centers were employed as supervisors to ensure there was no Logical error, wrong item or absent item in the questionnaire.” (Line 70-72)
3. 167- Instead of "old" consider the term "elderly."
Response: We agree that “elderly” is more appropriate and have replaced “old” with “elderly”.
4.197- "study objects" should be "study participants."
Response: We agree that “study participants” is more appropriate and have replaced “study objects” with “study participants”.
5. I believe that the authors did a nice job identifying the abilities and public confidence in primary care providers. My curiosity is in the definition of "hypertension" and the level of primary care use. Were the participants with mild hypertension more likely to seek primary care outside the hospital. While co-morbidity could signal this, it is less than direct.
Response: It is a good question. Participants would be marked as hypertension patients if they were diagnosed with hypertension before the surveillance (line80-81). However, the database can not provide the information of disease severity, and we can not recognize hypertension of different stages. We used morbidity to reflect severity indirectly. We agree that disease severity may influence choice of institutions for hypertension patients, and we put “morbidity” into the growth curve model to minimize the potential interruption. We have put it into the limitation part.
6. Did hypertension also include those suffering hypertensive urgency or hypertensive emergency? How effective were primary care physicians in treating (or referring) these patients?
Response: It is a good question and need further exploration. The surveillance recorded the utilization of outpatient for participants, but did not record detailed information. Whether there was any emergency situation and how was the outcome of treatment were both unknown.
7. Moderate English changes required
Response: We have paid high attention on the language. Before we submitted the manuscript, we have modified the language with the help of editing company named KGSupport (website: http://www.kgsupport.com/order512c.asp). And we have rechecked the language of revised manuscript by KGSupport.
Reviewer 2 Report
Thank you for the opportunity to review the manuscript: "Can Primary Medical Institutions Lead to Worse Health Status for Patients with Noncommunicable Diseases Compared with High-Level Hospitals? a Follow-up Observation Study in China"
The topic is attractive.
The paper is readable but needs to be clarified in some aspects.
Introduction:
Epidemiological data needs an update, Authors describe problem relevance and the specific lack of information in this specific setting, but their explanation sounds confusing when they talk about non-communicable diseases, then costs, quality of life, services availability and finally, hypertension. those arguments need to be related together more strongly
materials and methods
Authors did not describe the study design. Sampling and inclusion criteria are not declared and clear. Data collecting tool is validated and well described, authors should report tool psychometric properties. Statistical analysis is well described and reasonable.
results
sintetic and clear. Tables are well described.
discussion:
appropriate and related to the background, authors need to describe study limits
conclusions
authors described research potential, but they should suggest future improvement.
Author Response
Response to reviewers’ comments 2
ijerph-479573
Can Primary Medical Institutions Lead to Worse Health Status for Patients with Noncommunicable Diseases Compared with High-Level Hospitals? a Follow-up Observation Study in China
Dear editor and reviewer,
Thanks for your incredible work for reviewing our manuscript. Firstly, we would like to express our gratitude to you for your valuable input. Secondly, we have made some revisions in accordance with your comments and advices, and would like to re-submit the revised manuscript.
Our responses to your comments are below:
1. Epidemiological data needs an update, Authors describe problem relevance and the specific lack of information in this specific setting, but their explanation sounds confusing when they talk about non-communicable diseases, then costs, quality of life, services availability and finally, hypertension. those arguments need to be related together more strongly.
Response: It is a good suggestion. We collected epidemiological data from the websites of WHO, World Bank and National Health Department of China. we have rechecked those websites and found few updated reports.
We agree that the introduction was a little disorganized and have revised it.
The main idea of first paragraph: Non-communicable diseases is now a huge health threat and burden around the world. The current status of NCDs in China is no better than that in other countries.
The main idea of second paragraph: Chinese government wants to reduce the burden caused by NCDs through shifting the general health need of NCDs patients from hospitals to primary medical institutions. Under such policy, the health status of NCDs may suffer because of the weak primary health system and it needs evidence.
The main idea of third paragraph: Present study can not answer the question we put out.
The main idea of fourth paragraph: Brief description about how we answer the question in the second paragraph.
Revised introduction is as below:
Non-communicable diseases (NCDs) have become the major health threat in low-, middle-and high-income countries[1-3], and they account for more than 50% of global disease burden[4,5]. NCDs are also responsible for 70% of deaths worldwide, more than 40% of these deaths are premature ones[3]. Aging and changed lifestyle can make the status worse[6,7]. In 2030, over three quarters of all deaths will be caused by NCDs[1,3]. NCDs can not only influence individual’s life, but also impede sustainable development[1,8-10], thereby attracting attention from high-level politicians[8,10]. The current status of NCDs in China is no better than that in other countries. According to the Fifth Chinese National Health Services Survey, conducted in 2013, the prevalence rate of NCDs in China is 24.5% (much higher than the 18.9% in 2008), and 78.4% of the aged (over 65 years) presents at least one NCDs[11]. The World Bank and World health Organization indicate that NCDs are responsible for approximately 70% of the total disease burden and 89% of death in China[1,12].
Patients with NCDs present a large demand for health service, which consume a large amount of health resource[6,13]. Under such situation, the Chinese government hopes that the health need of patients with NCDs at stable or rehabilitation stage can be provided by primary medical institutions, and that patients with NCDs can seek for medical help in primary medical institutions during stable or rehabilitation period[14,15]. However, China has a weak primary healthcare system[16,17]. If patients with NCDs go to primary medical institutions for common health need, then whether their health will suffer compared with going to high-level hospitals is yet to be known.
To date, much study has been performed on factors influencing the health status of patients with NCDs[18-26]. These studies mainly focused on psychiatric condition[21,22,24,25], disease progression[18,20-22,24], comorbidities[23,25], health-risk behavior[19,26], and demographic characteristics[18,20,24,26]. Few studies have focused on the level of providers.
In this study, we selected out patients who seek for health care only in hospitals and those only in community health centers (or primary medical institution) to analyze the influence of provider level on the health status of patients with NCDs.
(Line 30-55)
2. Authors did not describe the study design. Sampling and inclusion criteria are not declared and clear. Data collecting tool is validated and well described, authors should report tool psychometric properties. Statistical analysis is well described and reasonable.
Response: We have modified the part of “Materials and Methods”. Line 58 descripted the study design. Line 61-62 declared the sampling criteria. Line 76-86 descripted the inclusion criteria and process. The Chinese version of EQ-5D-3L scale was used in the study, and the tool is widely used and tool psychometric properties has been reported by other researchers. (line 69-70)
Line 58: This is a follow-up observation study.
Line 61-62: Participants were selected from the samples of the Fifth National Health Service Survey of China in 2013, which was a Multi-stage stratified cluster random sampling.
Line 76-86:
Hypertension is a considerable risk factor of cardiovascular, diabetes and other chronic diseases[31]. It shows the highest morbidity among NCDs in China (43.1% of Patients with NCDs present hypertension), accounts for nearly 24% of outpatient service (the 2nd highest), and 5.4% of inpatient service (the 3rd highest)[11]. Therefore, we selected hypertension patients as study participants. Among the 936 respondents, 240 people were diagnosed with hypertension before, 126 of them used outpatient service for hypertension or relative disease during six months, and 107 of them filled out the EQ-5D completely by themselves for six times. We divided those 107 people into the following groups according to the institution they chose for outpatient service: group 1 only went to community health centers (49 people), group 2 only went to hospitals (43 people), and group 3 went to both community health centers and hospitals (15 people). Groups 1 and 2 were study participants, and we compared the trajectories of their HRQoL in six months.
Line 69-70:
It was measured by the Chinese version of EQ-5D-3L scale (EQ-5D) monthly; this measure is widely used and highly recommended[28-30]
3. authors need to describe study limits
Response: we have added the limitation part. (line 204-207)
Our study has several limitations. The sample size was small and the tracking time was short, which may influence the stability of conclusions. Only hypertension patients were included, and it may limit the extension of conclusions. Comorbidity can not fully reflect disease severity for hypertension patients and may cause bias of conclusions.
4. Authors described research potential, but they should suggest future improvement.
Response: we have suggested future improvement.(line 207-209)
Further studies need to include more disease, more participants with longer observation to enhance the stability and extension of conclusions. Besides, more characteristics of participants need to collected to reduce potential interruptions.
5. English language and style are fine/minor spell check required
Response: We have paid high attention on the language. Before we submitted the manuscript, we have modified the language with the help of editing company named KGSupport (website: http://www.kgsupport.com/order512c.asp). And we have rechecked the language of revised manuscript by KGSupport.